# Genome-Wide Analysis and Expression Profiling of the Glutathione Peroxidase-like Enzyme Gene Family in *Solanum tuberosum*

**DOI:** 10.3390/ijms241311078

**Published:** 2023-07-04

**Authors:** Shenglan Wang, Xinxin Sun, Xinyue Miao, Fangyu Mo, Tong Liu, Yue Chen

**Affiliations:** State Key Laboratory of Crop Stress Biology for Arid Areas, College of Agronomy, Northwest A&F University, Xianyang 712100, China; 2021055031@nwafu.edu.cn (S.W.); 2021055021@nwafu.edu.cn (X.S.); 2022055128@nwafu.edu.cn (X.M.); 2020055057@nwafu.edu.cn (F.M.); 2020055066@nwafu.edu.cn (T.L.)

**Keywords:** potato, glutathione peroxidase-like enzyme (GPXL), stress, gene expression

## Abstract

Glutathione peroxidase-like enzyme is an important enzymatic antioxidant in plants. It is involved in scavenging reactive oxygen species, which can effectively prevent oxidative damage and improve resistance. GPXL has been studied in many plants but has not been reported in potatoes, the world’s fourth-largest food crop. This study identified eight StGPXL genes in potatoes for the first time through genome-wide bioinformatics analysis and further studied the expression patterns of these genes using qRT-PCR. The results showed that the expression of StGPXL1 was significantly upregulated under high-temperature stress, indicating its involvement in potato defense against high-temperature stress, while the expression levels of StGPXL4 and StGPXL5 were significantly downregulated. The expression of StGPXL1, StGPXL2, StGPXL3, and StGPXL6 was significantly upregulated under drought stress, indicating their involvement in potato defense against drought stress. After MeJA hormone treatment, the expression level of StGPXL6 was significantly upregulated, indicating its involvement in the chemical defense mechanism of potatoes. The expression of all StGPXL genes is inhibited under biotic stress, which indicates that GPXL is a multifunctional gene family, which may endow plants with resistance to various stresses. This study will help deepen the understanding of the function of the potato GPXL gene family, provide comprehensive information for the further analysis of the molecular function of the potato GPXL gene family as well as a theoretical basis for potato molecular breeding.

## 1. Introduction

Reactive oxygen species (ROS) including oxygen ions, peroxides, and oxygen-containing free radicals are natural by-products of cell metabolism and are involved in cell signal transduction, growth, and development [1]. However, unfavorable environmental conditions can increase ROS production in plants. When the concentration of ROS exceeds the normal range, it can cause damage to cellular structures, leading to serious consequences known as oxidative stress. Thus, regulating the steady-state level of intracellular ROS is essential [2]. Oxidative stress in plants can induce the production of various enzymes including catalase (CAT), peroxidase (POD), superoxide dismutase (SOD), ascorbate peroxidase (APX), and glutathione peroxidases (GPXs) to reduce the production of reactive oxygen species. Among these enzymes, glutathione peroxidase (GPX) is a thiol peroxidase that plays a crucial role in eliminating oxygen free radicals. GPX utilizes the process of converting reduced glutathione (GSH) to oxidized glutathione (GSSG) to reduce hydrogen peroxide or organic hydroperoxides to water or the corresponding alcohols. This process prevents damage to cellular mechanisms. Alternatively, GPX can use reduced thioredoxin (Trx) for regeneration [3]. According to the substrates and the electron donor, all GPXs are classified as phospholipid hydroperoxide GPX, TRX peroxidases, or GPX-type enzymes [4]. The GPX homologs in plants are more inclined toward the TRX system rather than GSH as the electron donor [4]. Compared with mammals, the active site of plant glutathione peroxidase contains cysteine instead of selenocysteine [5]. To avoid confusion resulting from protein names that are based on homology and thus misleadingly suggest a functional link to glutathione, Safira Attacha et al., 2017 [4] studied GPX-homologs in *Arabidopsis* that were not able to use GSH as an electron donor and used GPX-like enzymes to name them. Therefore, our study in potatoes also used GPX-like enzymes (GPXL; EC 1.11.1.9) to name them.

So far, the GPXL gene family has been explored in many plants such as *Arabidopsis thaliana* [4], *Oryza sativa* [6], *Sorghum bicolor* [7], Zea mays, *Brachypodium distachyon* [1], Gossypium hirsutum [8], Panax ginseng [9], Pinus tabulaeformis [10], *Solanum lycopersicum* [11], Cucumis sativus [12], Citrus sinensis [13], etc. Numerous studies have demonstrated the involvement of the GPXL gene in stress response, highlighting its crucial role in safeguarding plants against diverse biotic and abiotic stressors including oxidative stress, drought, extreme temperatures, salinity, metal toxicity, and pathogen attacks [14]. The AtGPXL3 mutant in *Arabidopsis thaliana* exhibits heightened sensitivity to H_2_O_2_ during the processes of seed germination and seedling development [15]. The absence of AtGPXL1 and AtGPXL7 reduces the plant’s tolerance to photooxidative stress [16], and knocking out AtGPXL8 leads to a higher sensitivity of the plant to oxidative damage during root elongation [17]. Two GPXL genes were isolated and identified from the embryogenic callus of Panax ginseng. During salt and low-temperature stress, the expression of these two GPXL genes in Panax ginseng seedlings was significantly upregulated. However, in response to biotic stress, the expression of PgGPXL2 was inhibited [9]. The sorghum genome harbors seven GPXL genes, among which all except SbGPXL6 and SbGPXL7 exhibit upregulation in response to drought stress [7]. Following both cold stress and salt stress treatments, the rice GPXL gene exhibits upregulation [18], and OsGPXL3 can achieve tolerance to salt stress through an independent ROS clearance mechanism [19]. However, the rice GPXL gene demonstrates a downward expression trend after drought stress treatment [18]. The activity of glutathione peroxidase-like enzyme in barley root tips was considerably enhanced following Cu treatment and slightly increased after Pb, Ni, and Zn treatments [20]. This indicates that the GPXL gene exhibits differences and complexity in stress response between different species or different members of the same species. Furthermore, the GPXL gene family plays a pivotal role not only in plant defense, but also in the processes of plant growth, development, and signal transduction. Loss of AtGPXL1 and AtGPXL7 in *Arabidopsis thaliana* can result in morphological alterations in leaf cells and chloroplasts, influencing leaf development, light adaptation, and programmed cell death [16]. Silencing OsGPXL1 in rice mitochondria resulted in shortened stem length and reduced seed count in plants [21]. AtGPXL3 acts as an oxidative signaling transducer to specifically transmit H_2_O_2_ signals under ABA and drought stress [15]. OsGPXL3 is expressed 5-fold higher after ABA treatment and plays a role in ABA signaling [22].

The potato, being the fourth largest food crop globally, occupies a vast planting area exceeding 19 million hectares. The yield of potatoes is closely influenced by environmental conditions. While the GPXL gene exhibits effective responsiveness to abiotic and biotic stress, the potato GPXL gene family has lacked in-depth investigation to date. Hence, this study aimed to investigate the potato GPXL gene family and identify eight distinct potato GPXL genes. The physical and chemical properties, phylogenetic evolution, chromosome location, tandem replicated genes, gene structure, conservative motifs, cis-acting elements, protein interaction network, interspecies collinearity, GO annotation, gene expression, and qRT-PCR analysis of potato StGPXL protein were subjected to bioinformatics analysis methods. These analyses provided valuable insights for further elucidating the functions of the potato GPXL gene family.

## 2. Results

### 2.1. Identification of Members of GPXL Gene Family in Potato

Using the *Arabidopsis thaliana* and *Oryza sativa* GPXL gene family protein sequences, a Blastp search was carried out in the potato genome database, and eight potato protein sequences were screened. Simultaneously, using SPDE2.0 software and the GPXL gene domain (PF00255) to search for GPXL candidate genes in the potato protein sequence database, 10 sequences were found, and eight sequences were screened based on the E-value, which was consistent with the previous eight sequences. To verify the correctness of the preliminary identification results, we manually confirmed each sequence’s gene domain using the SMART and NCBI CDD websites. Finally, we identified eight GPXL genes from the potato genome (Table 1).

Table 1 provides detailed information about the eight GPXL genes including gene names, gene IDs, chromosome locations, open reading frame (ORF) length, exon numbers, protein lengths, molecular weights, predicted pI values, and subcellular localization predictions. The ORF size of the StGPXL proteins ranged from 303 bp to 2358 bp. The length of the proteins was 170 to 785 amino acids, the molecular weight was 19.26–87.58 kDa, and the predicted pI value was 4.77–10.21.

### 2.2. Phylogenetic Analysis of GPXL Gene Family in Potato

Multiple sequence alignment of GPXL proteins from potatoes with GPXL proteins from *Arabidopsis* was conducted. The results of the multiple sequence alignment (Figure 1) showed that multiple highly conserved characteristic domains were found in all GPXL protein sequences such as FTVKD, GKVLLIVNVASKCGLT, ILAFPCNQF, (I/L/V) F (D/E/Q/N) K (I/V) (D/E/R) VN (C/G), KWNF (A/E/T/S) KFLV, and Cys residues. Among them were GKVLLIVNVASKCGLT, ILAFPCNQF, and KWNF (A/E/T/S) KFLV. These three structural domains are characteristic motifs of GPXL. Among them, NVASKCGLT is the catalytic activity region of GPXL. However, in StGPXL2, StGPXL4, and StGPXL5, there are large mutations in this region, and there are more replacements and deletions, which may cause large differences in the functions of these genes and other gene family members. NVASKCG is conservative in the remaining five sequences, and K is the variable site.

To assess the evolutionary relationship between potato GPXL proteins, a neighbor-joining tree (NJ) was constructed using MEGA 11.0 software, and phylogenetic analysis was performed. In addition to potato GPXL proteins, the phylogenetic tree included GPXL proteins from model dicots (*Arabidopsis thaliana* [6], *Manihot esculenta Crantz*, *Solanum lycopersicum* [11], *Camellia sinensis*, *Nicotiana attenuata*), monocots (*Oryza sativa* [6], *Zea mays* L. [1], *Brachypodium distachyon* [1], *Sorghum bicolor* [7]), and gymnosperm *Pinus tabulaeformis Carr* [10] (Table A1). As shown in Figure 2, all potato GPXL genes were aggregated in the same branch as the GPXL genes in dicots. In the minimal polymeric branch, all of the potato GPXL genes were aggregated with the GPXL genes of tomatoes or tobacco, indicating that potatoes are more closely related to tomatoes and tobacco. StGPXL2, StGPXL4, and StGPXL5 were far from the genetic relatives of the other five GPXL genes, and a large degree of variation occurred, which was consistent with the results of the multiple sequence alignment.

### 2.3. Chromosomal Location and Tandem Duplication Genes of Potato GPXL Gene Family

The chromosome mapping results showed that eight potato GPXL genes were randomly and unevenly distributed on chromosomes 6, 8, and 12. There were two GPXL genes on chromosome 6, four GPXL genes on chromosome 8, and two GPXL genes on chromosome 12, respectively (Figure 3). There were replication events and high retention rates in the plant genome to the point that duplicate genes were present. Duplicate genes help evolve new functions. There are many amplification mechanisms of gene families including polyploidy, fragment replication, tandem duplication, transposable elements, etc. [23]. The members of a gene family formed by tandem duplication are usually closely arranged on the same chromosome, forming a gene cluster with similar sequences and functions. According to the defined criteria, two tandem repeats (StGPXL3, StGPXL4) were identified on chromosome 8, and two tandem repeats (StGPXL7, StGPXL8) were identified on chromosome 12.

From the perspective of the StGPXL gene structure (Figure 4), the number of exons of the StGPXL gene family members was between five and seven, except that StGPXL2 had four introns and StGPXL6 had six introns; the rest of the gene family contained five introns. From the gene structure map, the structural differences between the StGPXL gene family members were insignificant, indicating that the original structure of the StGPXL gene was not complicated, and a small number of gene family members were mutated.

### 2.4. Conserved Motif Identification

In order to further study the conservation and difference of the potato GPXL protein sequences, the conserved motifs of the potato GPXL protein sequences were analyzed using the MEME web page, and four conserved motifs were finally determined. Motif1 and motif2 had 50 amino acid lengths, motif3 had 41 amino acids in length, and motif4 had nine amino acids in length (Figure 5, Table 2). The results showed that the potato GPXL gene family contained inconsistent conserved motifs. GPXL1 contained motif1, motif2, and motif3; GPXL2 contained motif3 and motif4; GPXL4 and GPXL5 contained only motif4; the remaining four GPXL proteins contained four motifs. According to the Pfam website results, motif1 and motif2 both belonged to the GSHPx domain (PF00255). In the multi-sequence alignment results, the conservative sequences ILAFPCNQF and IFDKIDVNG belonged to motif1; FTVKD and GKVLLIVNVASKCGLT belonged to motif2; KWNFAKFLV belonged to motif3. Mapping using SPDE2.0 software (Figure 6), the GPXL4 and GPXL5 proteins still contained GSHPx domains. The motif is closely related to biological function, and different motif structures between different genes in the same gene family indicate the diversity of their functions.

### 2.5. Cis-Acting Elements Analysis

In order to further study the response of the StGPXL gene to abiotic stress, the 2001 bp sequence upstream of each gene translation initiation site was analyzed, and a variety of cis-acting elements were found. According to the functions of cis-acting elements, 21 cis-acting elements related to plant growth and development, hormone response, and environmental stress response were screened (Figure 7). The most StGPXL2 had 16 cis-acting elements, and the least StGPXL7 and StGPXL8 had five cis-acting elements. There were seven stress response elements as follows: light induction (G-box, G-Box, I-box); low-temperature response (LTR); high-temperature response (AT-rich-sequence, STRE); drought response (MBS); anaerobic induction (ARE); mechanical damage (WUN-motif); adversity defense response (TC-rich-repeats). There were six hormone-responsive elements as follows: abscisic acid (ABRE); methyl jasmonate (G-box, G-Box, TGACG-motif, CGTCA-motif, W-box); gibberellin (TATC-box, P-box); ethylene (ERE); auxin (TGA-element); salicylic acid (TCA-element). Two cis-acting elements related to growth and development regulation were as follows: meristem specificity (CAT-box) and circadian rhythm regulation (circadian). Among the cis-acting elements, abscisic acid (ABRE) and ethylene (ERE) had the highest numbers, 25 and 21, respectively. High-temperature response elements (STRE) were contained in each gene, while the adversity defense response elements (TC-rich-repeats) and auxin response elements (TGA-element) were only contained in StGPXL6. The analysis of cis-acting elements showed that the StGPXL gene family was closely related to plant abiotic stress, growth and development, and hormone regulation.

### 2.6. Potato GPXL Protein Interaction Analysis

Using *Arabidopsis thaliana* as a reference species, the interacting proteins of the StGPXL gene family were predicted by the STRING protein database. The results showed that (Figure 8) the homologous protein of StGPXL1 was AtGPXL2; the homologous protein of StGPXL2 was AtGPXL5; the homologous protein of StGPXL3 was AtGPXL7; the homologous protein of StGPXL5 was AtGPXL1; the homologous protein of StGPXL6 was AtGPXL6; the homologous protein of StGPXL7 and StGPXL8 was AtGPXL8. However, StGPXL4 did not find a homologous protein in Arabidopsis. The proteins that interact with the GPXL protein are mainly APX, CAT, CAT1, F5M15.5 (AT1G20620, CAT3), DHAR, GR, AT3G24170 (GR1), and GSH2. Catalase (CAT) is commonly present in plants and is used to clear ROS. The ascorbate–glutathione cycle is an important way to eliminate ROS in plants, which exists in a variety of subcellular organelles such as chloroplasts, mitochondria, cytoplasm, etc. [24]. The ascorbate–glutathione cycle specifically includes ascorbate peroxidase (APX), dehydroascorbate reductase (DHAR), glutathione reductase (GR), monodehydroascorbate reductase (MDHAR), ascorbic acid (AsA), and reduced glutathione (GSH). APX catalyzes the reaction of AsA with H_2_O_2_ to produce monodehydroascorbate (MDHA) and H_2_O, thereby clearing H_2_O_2_ produced under stress [25]. MDHA has a short half-life and can quickly be disproportionately reduced to AsA or oxidized to dehydroascorbic acid (DHA) [26]. MDHAR catalyzes the reduction of MDHA by NADH to produce AsA and NAD^+^. DHAR catalyzes the reduction of DHA by GSH to produce AsA and oxidized glutathione (GSSG), regulating the cell AsA/DHA ratio, while GR catalyzes GSSG to GSH, thereby completing the process of scavenging reactive oxygen species and regenerating AsA and GSH [27]. Therefore, there is an interaction between the AsA–GSH cycle in plants and the metabolic activity of GSH. Glutathione synthase (GSH2) catalyzes the early biosynthesis of glutathione [28]. GSH is the substrate of GPXL. GPXL and GR jointly form the glutathione peroxidase cycle, causing mutual transformation between GSH and GSSG [24]. GPXL forms an interaction network with the various enzymes above-mentioned, which together play a role in clearing ROS.

### 2.7. Collinear Analysis of GPXL Gene in Potato and Arabidopsis thaliana

From the collinear analysis of the genes, there were six pairs of homologous genes in the potato GPXL gene and the Arabidopsis GPXL gene as follows: StGPXL1 and AtGPXL3; StGPXL3 and AtGPXL1; StGPXL3 and AtGPXL7; StGPXL6 and AtGPXL6; StGPXL7 and AtGPXL6; StGPXL7 and AtGPXL8 (Figure 9). This showed that there was a homologous evolutionary relationship between the potato GPXL gene family and the Arabidopsis GPXL gene family, and the four potato GPXL genes of StGPXL1, StGPXL3, StGPXL6, and StGPXL7 played an important role in the evolution. In addition, StGPXL2, StGPXL4, StGPXL5, and StGPXL8 did not have collinearity genes in Arabidopsis, indicating that these four GPXL genes are specific genes during potato evolution.

### 2.8. Gene Ontology Annotation of StGPXL Proteins

In order to further study the biological process and molecular function of the StGPXL gene family proteins, due to the lack of annotation information, only StGPXL3, StGPXL7, and StGPXL8 were finally analyzed by GO annotation (Figure 10). The biological processes of all three proteins were responses to oxidative stress. The molecular functions of the three proteins were peroxidase activity and glutathione peroxidase activity. For cellular components, however, only StGPXL7 and StGPXL8 were in the cytosol.

### 2.9. Organ Expression and Stress Treatment Expression Analysis of the Potato GPXL Genes

In order to study the expression of potato GPXL genes in callus and various other plant organs during growth and development as well as the changes in the expression level of the potato GPXL gene during stress response, RNA-Seq data can be obtained from genetic databases, and heat maps can be drawn. The expression levels of the potato GPXL genes in callus, carpels, flowers, leaves, petals, petioles, roots, shoots, stamens, stems, stolons, and tubers (Figure 11, Table A2) and the expression levels of the potato GPXL genes under abscisic acid (ABA), 6-benzylaminopurine (BAP), gibberellic acid (GA3), auxin (IAA), heat, mannitol, salt, β-aminobutyric acid (BABA), benzothiadiazole (BTH), and *P. infestans* (Figure 12, Table A3) were demonstrated.

The expression results of the potato GPXL gene family in various organs and callus showed that StGPXL1, StGPXL6, and StGPXL8 were expressed to a high degree in all tested organs and callus, and the expression of StGPXL1 reached its peak in the stamens, StGPXL6 in the callus, and StGPXL8 in the stolons. These findings underscore the pivotal regulatory roles played by these three genes in the growth and development of potatoes. In contrast, StGPXL4 and StGPXL5 showed extremely low or no expression levels in all of the tested organs. The expression level of StGPXL4 was deficient in the stems and stolons, while the expression level of StGPXL5 was low in the petioles, stems, and stolons. In addition, all GPXL genes were expressed in the stolons, indicating a close relationship between the GPXL genes and the growth and development of the stolons. The functions of different StGPXL in various organs of the potato were different. The expression results of the potato GPXL gene under various stress conditions showed that all genes except StGPXL4 and StGPXL5 were upregulated. StGPXL6 was upregulated and reached its peak under the action of BABA.

### 2.10. Expression Analysis of StGPXL Genes in Different Treatments

Based on the above bioinformatics analysis, in order to further study the response of the potato GPXL gene to abiotic stress and biotic stress, and combined with the analysis of cis-acting elements, potato material Desiree was treated with high temperature, drought, methyl jasmonic acid, and *Ralstonia solanacearum* GMI1000, and the expression level of the StGPXL gene was quantitatively analyzed by qRT-PCR (Figure 13). The results showed that all four types of stress changed the expression levels of the GPXL genes in all potatoes, and the degree of change varied due to different stresses and genes.

As shown in Figure 13A, the changes in the gene expression levels under 38 °C treatment were consistent with the RNA-Seq data. Except for the StGPXL4 and StGPXL5 genes, which were significantly downregulated by 0.71-fold and extremely significantly downregulated by 0.83-fold, all of the other six genes showed upregulation changes. StGPXL1 was significantly upregulated by 0.79 times. Under drought treatment (Figure 13B and Figure 14), all genes underwent upregulation changes, with StGPXL1, StGPXL2, StGPXL3, and StGPXL6 showing particularly significant upregulation of 128.44-, 13.71-, 12.84-, and 84.16-fold, respectively. Under methyl jasmonate treatment (Figure 13C and Figure 14), except for StGPXL5 and StGPXL7, which were slightly insignificant downregulated changes, the other genes underwent upregulated changes. Among them, StGPXL6 significantly increased by 46.89 times. However, the expression of all genes treated with GMI 1000 was insignificantly inhibited (Figure 13D and Figure 14).

## 3. Discussion

Unfavorable climatic circumstances might cause plants to produce an excessive amount of ROS. Glutathione peroxidase-like enzyme is an essential antioxidant enzyme in plants, preventing oxidative damage and improving resistance [12]. This study identified eight GPXL genes in the potato genome, with a number approximate to that of GPXL genes in plants such as Arabidopsis (eight members) [6], sorghum (seven members) [7], cucumber (six members) [12], and castor (five members) [29]. Therefore, there was no significant amplification of the potato GPXL gene during the evolutionary process. In this study, we found two tandem duplication events (StGPXL3 and StGPXL4; StGPXL7 and StGPXL8), similar to those in castor (RcGPXL4 and RcGPXL5) [29] and cucumber (CsGPXL3 and CsGPXL4) [12]. The presence of tandem repeat genes can promote genetic innovation and evolution, and increase the diversity of gene expression. In the study of gene structure, there are five to seven exons of the potato GPXL gene, eight exons of the Arabidopsis GPXL gene, and five to six exons of the castor GPXL gene [29]. Therefore, the GPXL gene family is highly conservative in the arrangement of introns and exons among different plants, which plays a key role in the gene family’s evolution and makes the gene family have similar functions. Multiple sequence alignment and domain mapping showed that all eight potato GPXL genes contained complete GPXL conserved domains and multiple conserved protein sequences. Notably, the enzymatic activity of plant GPXL differs from that of animals, as plant GPXL relies on the first two out of three crucial cysteine residues [3,4] while the cysteine residues in StGPXL2, StGPXL4, and StGPXL5 are missing, leading to the functional diversity of the GPXL gene family, which is related to the low expression of these three genes in subsequent plant organs and under stress conditions (Figure 11 and Figure 12). In addition, the potato GPXL gene contains one to four conserved motifs, constituting the main conserved functional region of the StGPXL protein. The eight GPXL genes of the potato are distributed on chromosomes 6, 8, and 12, and there are six pairs of collinear genes from Arabidopsis. Collinear genes are closely related in phylogenetic trees, and the potato is closely related to two solanaceous plants (tobacco and tomato). Moreover, genes with consistent subcellular localization predictions are more likely to cluster on the same branch, which is closely related to gene evolution. Most of the protein localization of the StGPXL gene expression is located in chloroplasts due to PSI being one of the main sources of H_2_O_2_ production in plants as well as the involvement of the AsA–GSH cycle in the water–water cycle in chloroplasts [30,31]. Arabidopsis genes with collinearity with StGPXL have been reported many times; for example, AtGPXL controls the root structure by mediating redox regulation [32]; the expression levels of AtGPXL2 and AtGPXL7 increase under ultraviolet (UV-B) stress to eliminate the lethal damage of ultraviolet radiation to plants [33]; AtGPXL3 not only acts as a scavenger under ABA and drought stress but also as a signaling molecule to specifically transmit H_2_O_2_ signals [15]. At the same time, it can physically interact with SRO1 to regulate the oxidative homeostasis in ROS cells under heavy metal mercury stress to avoid plant damage [34]; AtGPXL1 and AtGPXL7 affect leaf development and tolerance to photooxidative stress in Arabidopsis [16]. Therefore, it can be inferred that the StGPXL gene will have similar biological functions.

The gene expression analysis in potato organs showed that the StGPXL gene family was widely expressed in various organs. However, the expression was different, and all StGPXL genes were expressed in the stolons. Stolons are a necessary condition for potato tuber formation as well as the foundation and prerequisite for potato yield. Therefore, the StGPXL gene plays a vital role in clearing ROS produced by potato stolon cell metabolism.

Through the analysis of protein interaction networks, glutathione peroxidase is involved in the glutathione peroxidase cycle and the ascorbate–glutathione cycle. It is one of the crucial enzymes for scavenging ROS. In addition, RNA-Seq data in the potato genome database and qRT-PCR experiments found that the StGPXL gene responded to various stresses, consistent with the predicted cis-acting element function. The promoter region containing cis-acting elements is related to the gene expression of plant growth and stress adaptation. The potato GPXL promoter region contains many cis-acting elements related to abiotic stress and hormone response such as low-temperature response (LTR), high-temperature response (AT-rich-sequence, STRE), drought response (MBS), abscisic acid (ABRE), salicylic acid (TCA-element) (Figure 7), etc. However, the expression levels of different genes vary under different stresses. Plants are affected by various abiotic factors during their growth and development, with temperature being one of the main factors. High-temperature stress can cause stomatal closure, negatively affecting photosynthesis and respiration and a decrease in yield. In this study, the expression level of StGPXL1 significantly increased under high-temperature stress, achieving the scavenging effect of glutathione peroxidase-like enzyme on ROS. In previous studies, after 24 h of high-temperature treatment, *Gossypium hirsutum* significantly increased the expression levels of GhGPXL5 in the leaves, and GhGPXL1, GhGPXL2, GhGPXL3, GhGPXL5, GhGPXL8, and GhGPXL13 in the roots [8], indicating that the GPXL gene plays a vital role in the clearance of H_2_O_2_ under high-temperature stress. Among the eight StGPXL genes, each gene has a high-temperature cis-acting element (STRE). StGPXL4 and StGPXL5 showed a downward expression trend under high-temperature stress, consistent with the potato gene database results. The reasons for their downregulation can be further explored. Drought is another severe threat that plants face. When plants experience water stress, the concentration of endogenous ABA rapidly increases and acts as a signaling molecule to transmit stress signals, control stomatal opening, induce the expression of stress-related genes, and promote plant adaptation to stressful environments [35]. Overexpression of Arabidopsis AtGPXL3 not only regulates the homeostasis of H_2_O_2_, but also enhances plant drought tolerance through the ABA pathway mediated by Ca^2+^ channels [15]. Under drought stress, the accumulation of H_2_O_2_ in wild-type plants is higher than that in transgenic plants. Overexpression of RcGPXL5 upregulates the activity of ROS scavenging enzymes in plants, while stress induces an increase in the expression of the genes CBF and protein phosphatase 2C (ABI2) to increase plant drought resistance [36]. The expression of all StGPXL genes in potatoes is upregulated under drought stress conditions, with StGPXL1, StGPXL2, StGPXL3, and StGPXL6 being particularly significant. It can be inferred that there are genes that can mediate ABA. Exogenous application of jasmonic acid (JA) and methyl jasmonate (MeJA) can stimulate the expression of plant defense genes and produce a chemical defense, which is similar to mechanical injury and insect feeding [37]. After 3 h of treatment with JA in *Gossypium hirsutum*, the expression level of the GhGPXL8 gene in the leaves and roots was specifically increased [8]. Similar phenomena were observed in potatoes, where MeJA treatment specifically increased the expression level of the StGPXL6 gene. In ginseng, the expression levels of PgGPXL1 and PgGPXL2 decreased after JA treatment [9]. This can indicate that the GPXL gene has functional differentiation, and family members will play a role under different stress conditions. Infection by pathogens can trigger plant immune responses, and a rapid increase in intracellular reactive oxygen species levels is the first barrier to plant immunity. To investigate whether the StGPXL gene can clear ROS and maintain stable cell stability after biological stress, this experiment used *Ralstonia solanacearum* to infect potato roots as a biological stress condition and found that the expression of the StGPXL gene showed a downward trend. In ginseng, the expression of PgGPXL1 increased under biotic stress, while PgGPXL2 showed the opposite [9]. This indicates that the GPXL gene usually plays a role under abiotic stress and also confirms the diversity of its functions.

## 4. Materials and Methods

### 4.1. Plant Materials Preparation

In this study, the potato variety Desiree was used as the experimental material. The experiment was carried out at the State Key Laboratory of Crop Stress Biology in the Arid Zone, Northwest A&F University (107°59′–108°08′ east longitude, 34°14′–34°20′ north latitude). When performing tissue culture seedling cultivation, the medium used was Murashige and Skoog (MS) liquid medium containing 2% sucrose, 0.05% MES (2-morpholinoethanesulfonic acid), and pH 5.9 [38]. The study period was from December 2022 to January 2023. Tissue culture seedlings were incubated in an incubator at 22 °C, 16 h light (10,000 Lx), 8 h dark, and 70% relative humidity for three weeks. Then, the following treatments were performed: the tissue culture seedlings were divided into six groups, and the first group was incubated at 38 °C for 6 h; the second group transferred the tissue culture seedlings into a nutrient solution containing 20% polyethylene glycol (PEG4000) for 24 h. The third group transferred the tissue culture seedlings into a nutrient solution containing 10 μM methyl jasmonate, cultured for 3 h, and the fourth group was not treated as a control group; in the fifth group, the tissue culture seedlings were transferred into the bacteria solution containing GMI1000 (OD600 = 0.01) and cultured for 12 h (the bacteria solution was configured with dH_2_O and sterilized); for the sixth group, as the control group of the fifth group, we transferred the tissue culture seedlings into dH_2_O and incubated for 12 h. The first four groups collected potato leaves for RNA extraction. Groups 5 and 6 collected plant roots from potatoes for RNA extraction. Three biological replicates per treatment condition were set up to reduce the error rates [39].

### 4.2. Identification of GPXL Gene Family in Potato

Using the protein sequences of the GPXL gene family identified in the *Arabidopsis thaliana* (http://www.arabidopsis.org/index.jsp (accessed on 21 November 2022)) genome database as query sequences, the potato genome database was searched using local Blastp (http://solanaceae.plantbiology.msu.edu/blast.shtml (accessed on 22 November 2022)). The sequence information of the potato homologous StGPXL gene family members was obtained (Table A1). The domain (PF00255) model file of the GPXL gene family from the Pfam database (http://pfam.xfam.org/ (accessed on 22 November 2022)) was downloaded. SPDE2.0 software and HMMER 3.0 software were used to screen the potato protein sequence containing the GPXL domain. Then, to test whether the initial identification results were correct, we manually confirmed the gene domain of each sequence using the SMART (http://smart.embl-heidelberg.de/ (accessed on 22 November 2022)) and NCBI CDD (http://www.ncbi.nlm.nih.gov/Structure/cdd/wrpsb.cgi (accessed on 22 November 2022)) websites. Using the Expasy website (http://web.expasy.org/protparam/ (accessed on 25 November 2022)) and combined with the potato genome database information, we predicted and analyzed the physicochemical properties of all GPXL protein sequences [40].

### 4.3. Multiple Sequence Alignment and Phylogenetic Tree Construction

Multi-sequence comparison of the GPXL proteins in potatoes and *Arabidopsis thaliana* was conducted using Jalview to identify the domain and conserved domain residues. The ClustalW algorithm was applied to compare the potato GPXL family with *Arabidopsis*, cassava, tomato, maize, tea tree, *Nicotiana attenuata*, rice, *Brachypodium distachyon*, sorghum, and *Pinus tabulaeformis*, and the MEGA 11.0 software built a phylogeny tree with neighbor-joining, bootstrap replicates (1000), p-distance [41].

### 4.4. Chromosomal Location Analysis and Tandem Replicated Genes and Gene Structure

Potato gene location information was downloaded from the gff3 files of the potato genome database (http://solanaceae.plantbiology.msu.edu/ (accessed on 29 November 2022)). The chromosome distribution of the potato GPXL gene was analyzed and mapped by TBtools v1.108 software [42]. Two genes in the same family, with a sequence similarity of more than 70%, a gene interval within five genes, and a distance of less than 100 kb were defined as tandem replicated genes [43]. The storage file of the distribution of the exons and introns of the potato GPXL gene was downloaded from the potato genome database website, and TBtools software was used to draw the gene structure map.

### 4.5. Conserved Motif Identification and Cis-Acting Elements Analysis

The conserved motifs of the potato GPXL protein were analyzed using the MEME website [44] (http://meme-suite.org/ (accessed on 1 December 2022)) and plotted using the SPDE2.0 software. In order to study the cis-acting elements in the promoter region of the potato GPXL gene, the 2001 bp sequence before the start codon of the GPXL gene was retrieved from the potato genome database and submitted to the PlantCARE online website (http://bioinformatics.psb.ugent.be/webtools/plantcare/html/ (accessed on 2 December 2022)) to predict the cis-acting elements. Then, SPDE2.0 software was used to plot a distribution map of the cis-acting components in the promoter region.

### 4.6. Analysis of GPXL Protein Interaction in Potato

We uploaded each of the eight StGPXL proteins to the STRING website (http://cn.string-db.org/ (accessed on 5 December 2022)), selected *Arabidopsis thaliana* as a model plant to find homologous proteins, built a protein interaction network, and then used Cytoscape 3.10.0 software to plot the results.

### 4.7. Gene Ontology Annotation and Interspecific Collinearity Analysis of GPXL Gene in Potato

The Gene Ontology (GO) analysis of the potato GPXL protein was performed using the David website [45] (https://david.ncifcrf.gov/ (accessed on 7 December 2022)). After uploading the serial numbers of eight potato GPXL proteins in GENBANK, David’s website analyzed their biological processes, molecular functions, and cellular components. TBtools software was used to plot the GPXL gene collinearity map of potatoes and Arabidopsis.

### 4.8. Organ Expression and Stress Treatment Expression Analysis of the Potato GPXL Genes

To explore the potential role of StGPXL in growth and development, we downloaded the potato GPXL gene family for fragments per kilobase per million (FPKM) values from the PGSC database (http://solanaceae.plantbiology.msu.edu/dm_v6_1_download.shtml/ (accessed on 9 December 2022)). We calculated the value of Log2 after removing the value of less than one in organ and stress and used TBtools software to plot the expression heat map of different organs and GPXL gene families with different treatments [46].

### 4.9. RNA Isolation and qRT-PCR Analysis

The total RNA from the potato was extracted using the RNAsimple Total RNA Extraction Kit (TIANGEN, Beijing, China, DP419). cDNA was synthesized using the PerfectStart Uni RT&qPCR Kit (TRAN, Beijing, China, AUQ-01-V2), according to the manufacturer’s instructions. The specific potato GPXL gene primers for quantitative real-time PCR (qRT-PCR) were designed using Primer Premier 6 software and NCBI. The ef1α gene was used to serve as an internal control for normalized gene expression levels (Table A4). qRT-PCR was performed using the SuperReal PreMix Color (SYBR Green) reagent as well as a 20 μL reaction system selected with the following composition: 10 μL 2 × SuperReal Color PreMix, 0.75 μL 10 μM forward primer, 0.75 μL 10 μM reverse primer, 1.5 μL cDNA, 0.8 μL 50 × ROX Reference Dye, and 6.2 μL RNase-free ddH_2_O. The qRT-PCR process was performed on a Q7 Real-Time PCR System and the qRT-PCR program was set to: initial activation of 95 °C for 15 min, then 40 cycles of 95 °C for 10 s, 56 °C for 20 s, and 72 °C for 20 s. The relative expression levels were calculated using the comparative 2^−∆∆CT^ method [47].

## 5. Conclusions

This study identified eight StGPXL genes from potatoes for the first time, distributed on three chromosomes and containing the GSHPx domain. Phylogenetic tree analysis showed that the eight StGPXL genes were closely related to the GPXL gene family of tobacco and tomato. The analysis of cis-acting elements showed that StGPXL was involved in coping with and defending against potato damage under abiotic stress such as hormones and adverse environments. The qRT-PCR study showed that the expression level of StGPXL1 significantly increased and participated in the response to high-temperature stress, while the expression levels of StGPXL4 and StGPXL5 were inhibited. StGPXL1, StGPXL2, StGPXL3, and StGPXL6 jointly participate in the response to drought stress. StGPXL6 responds specifically to MeJA. All StGPXL genes were inhibited in expression under biotic stress. This indicates that GPXL is a multifunctional gene family, which can enable plants to effectively cope with a variety of abiotic stress. Identification of the potato GPXL gene family through the whole genome is a more comprehensive understanding of the diversity of this gene family. The protein interaction network revealed the potential value of the StGPXL protein. This study is intended to provide valuable clues for the subsequent function research of the gene family.

## Figures and Tables

**Figure 1 ijms-24-11078-f001:**
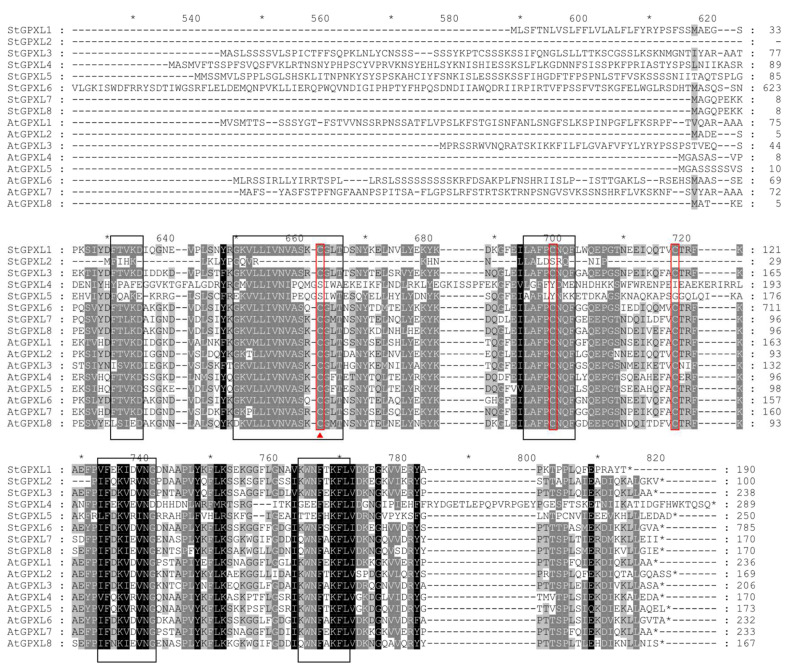
Multiple sequence alignment of the potato GPXL proteins and *Arabidopsis* GPXL proteins. The alignment was performed using MAFFT with defaults, followed by shading with conservation. The darker the color of the region, the more conservative it is. Black boxes indicate conservative domains. The red boxes represent three conserved Cys residues. The red triangles represent Cys residues in mammals that have been replaced by Sec.

**Figure 2 ijms-24-11078-f002:**
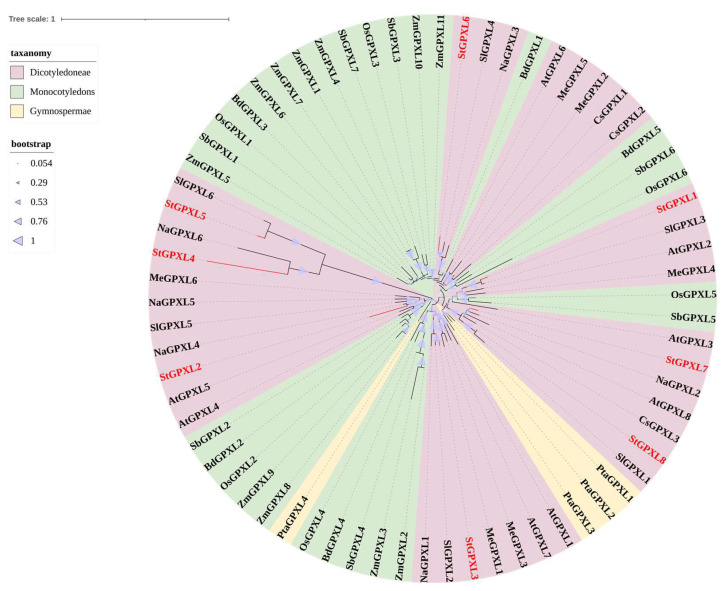
Phylogenetic analyses of the plant GPXL proteins. The conserved GPXL proteins from *Solanum tuberosum* L., *Arabidopsis thaliana*, *Manihot esculenta Crantz*, *Solanum lycopersicum*, *Zea mays* L., *Camellia sinensis*, *Nicotiana attenuata*, *Oryza sativa*, *Brachypodium distachyon*, *Sorghum bicolor*, and Pinus tabulaeformis Carr were aligned using the ClustalW function of MEGA11.0, and the phylogenetic tree was constructed using the Neighbor-Joining (NJ) method with bootstrapping analysis (1000 replicates).

**Figure 3 ijms-24-11078-f003:**
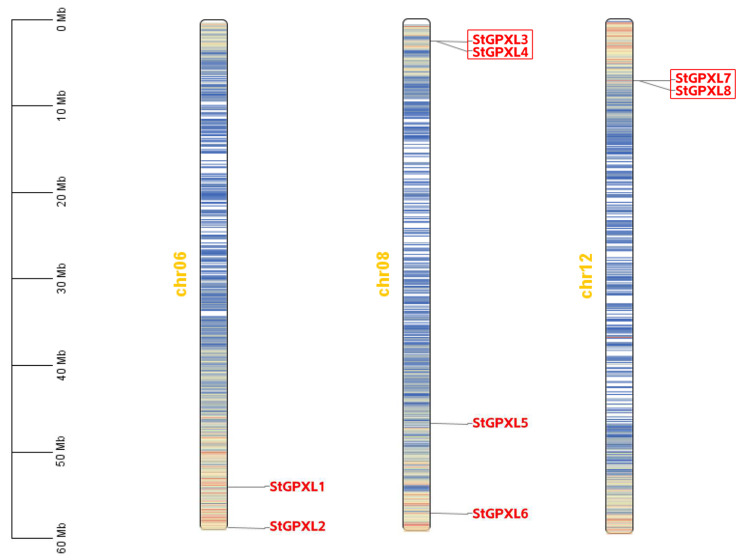
Chromosomal distribution of the potato GPXL genes. Red boxes indicate tandem duplicate genes. Blue lines on chromosomes indicate gene density.

**Figure 4 ijms-24-11078-f004:**
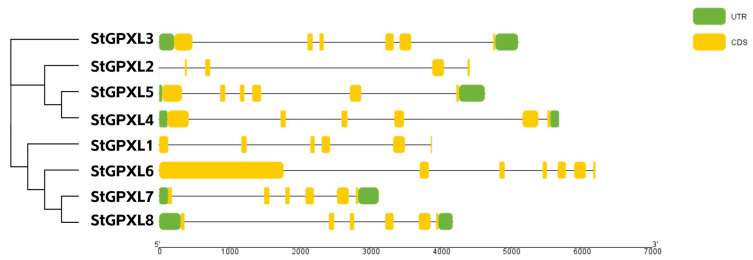
Gene structure of the potato GPXL genes. Yellow color bars represent the coding sequence (CDS), lines represent the intron, and green color bars represent the untranslated region (UTR).

**Figure 5 ijms-24-11078-f005:**
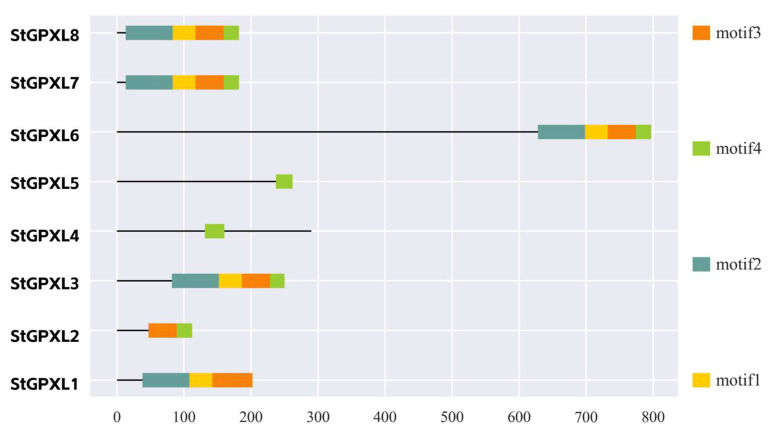
The conserved motif of the potato GPXL proteins. Different colors indicate different motifs.

**Figure 6 ijms-24-11078-f006:**
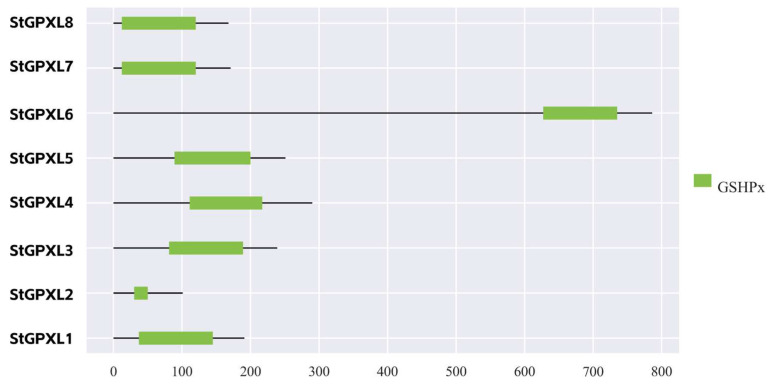
Domains of the potato GPXL proteins.

**Figure 7 ijms-24-11078-f007:**
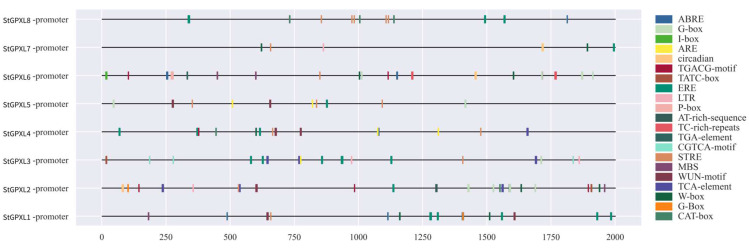
The cis-acting element of the 2001 bp sequence upstream of the potato StGPXL gene. This study used the database PlantCARE to predict the motif.

**Figure 8 ijms-24-11078-f008:**
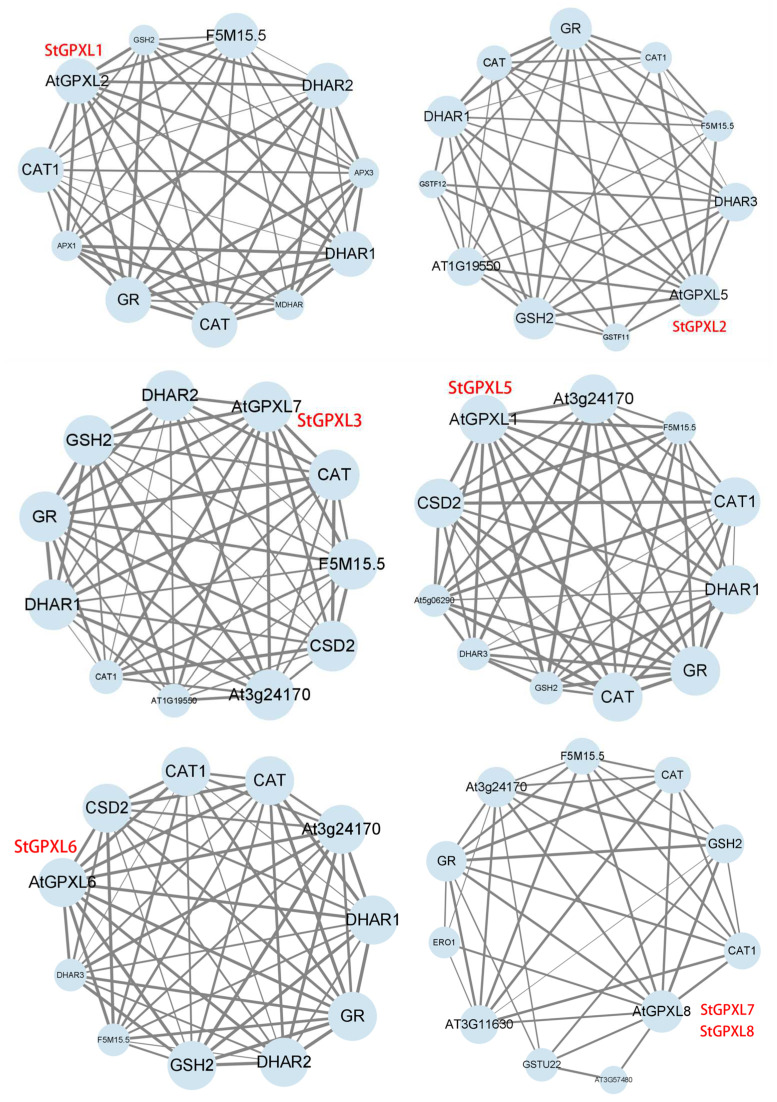
Prediction of the function network of the interacting protein of StGPXL. The size of the circle indicates how often the protein appears (the larger the area, the higher the frequency). The thickness of the line segment between proteins indicates the combined score of the correlation degree between two proteins (the thicker the line segment, the higher the score).

**Figure 9 ijms-24-11078-f009:**
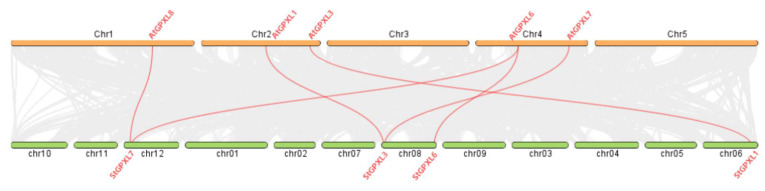
Homology analysis of the GPXL genes between potato and Arabidopsis. Gray lines: Collinear blocks between potato and Arabidopsis genomes. Red lines: Syntenic GPXL gene pairs between potato and Arabidopsis.

**Figure 10 ijms-24-11078-f010:**
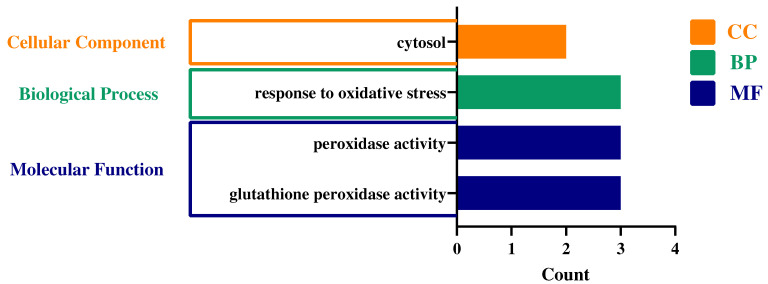
Gene Ontology (GO) annotation of the StGPXL3, StGPXL7, and StGPXL8 proteins.

**Figure 11 ijms-24-11078-f011:**
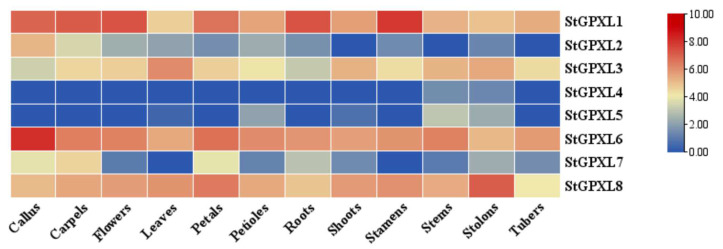
The heatmap shows the expression of the StGPXL gene in the organs or callus tissues including the callus, carpels, flowers, leaves, petals, petioles, roots, shoots, stamens, stems, stolons, and tubers. Red indicates high relative gene expression, whereas blue indicates low relative gene expression.

**Figure 12 ijms-24-11078-f012:**
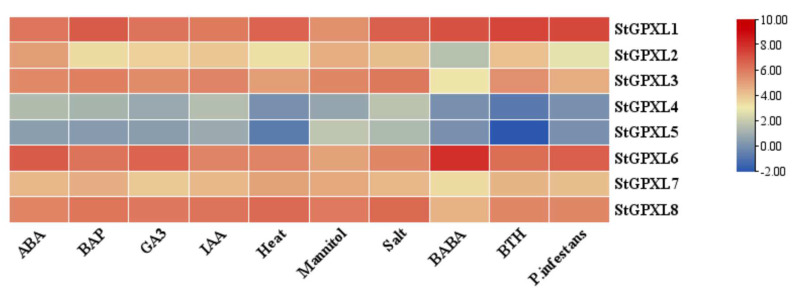
Heatmap of the expression profile of the potato GPXL genes under 10 different biotic or abiotic stresses. Biotic stresses included β-aminobutyric acid (BABA), benzothiadiazole (BTH), and *P. infestans*; abiotic stresses included heat, mannitol, and salt; other stress responses were mainly induced by the following four plant hormones: abscisic acid (ABA), 6-benzylaminopurine (BAP), gibberellic acid (GA3), and auxin (IAA). Red indicates gene upregulation, while blue indicates gene downregulation.

**Figure 13 ijms-24-11078-f013:**
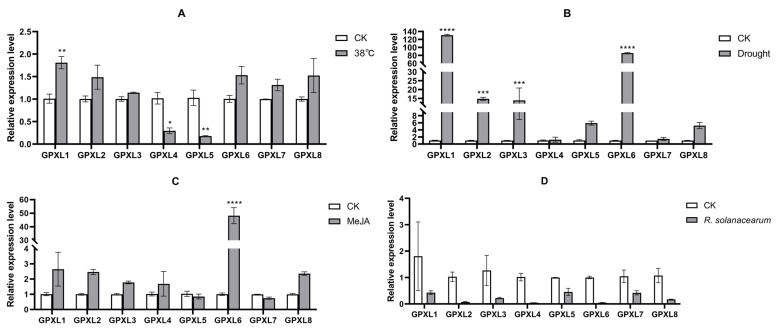
Differential expression of the GPXL gene family in response to abiotic stress, biotic stress, and hormone induction. (**A**) Relative expression level at 38 °C. (**B**) Relative expression level under drought treatment. (**C**) Relative expression level under MeJA treatment. (**D**) Relative expression level under conditions of *Ralstonia solanacearum* infection (* *t*-test *p*-value < 0.05, ** *t*-test *p*-value < 0.01, *** *t*-test *p*-value < 0.001, **** *t*-test *p*-value < 0.0001).

**Figure 14 ijms-24-11078-f014:**
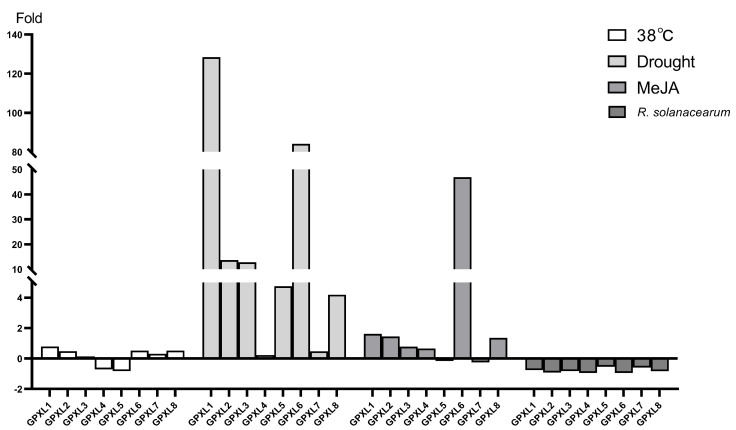
Fold changes in the expression level of the potato GPXL gene under different stresses.

**Table 1 ijms-24-11078-t001:** The GPXL genes in potatoes and the properties of the deduced proteins (*Solanum tuberosum*).

Gene ^1^	Gene ID ^1^	Chromosome Location (bp) ^1^	ORF Length (bp) ^1^	No. of Exons ^1^	Protein ^2^			Subcellular Location ^3^
					Length (aa)	MW (Da)	pI	
**StGPXL1**	Soltu.DM.06G028790.1	Chr06:54035324-54031449 (−)	573	6	190	21,786.10	8.30	Chloroplast
**StGPXL2**	Soltu.DM.06G034810.1	Chr06:58777923-58773510 (−)	303	5	100	11,227.20	10.21	Cytoplasm
**StGPXL3**	Soltu.DM.08G001820.1	Chr08:2523395-2528493 (+)	717	6	238	26,153.87	9.06	Chloroplast
**StGPXL4**	Soltu.DM.08G001830.1	Chr08:2533676-2539357 (+)	870	6	289	33,662.06	7.17	Chloroplast
**StGPXL5**	Soltu.DM.08G018060.1	Chr08:46713222-46708596 (−)	753	6	250	28,095.02	9.27	Nucleus
**StGPXL6**	Soltu.DM.08G027650.1	Chr08:57099704-57105902 (+)	2358	7	785	87,581.52	5.22	Endoplasmic reticulum
**StGPXL7**	Soltu.DM.12G008100.1	Chr12:7063794-7066913 (+)	513	6	170	19,475.17	4.77	Cytoplasm
**StGPXL8**	Soltu.DM.12G008110.1	Chr12:7070265-7074434 (+)	513	6	170	19,257.85	5.27	Cytoplasm

^1^ Gene information was retrieved from the *S. tuberosum* v6.1 genome annotation (http://solanaceae.plantbiology.msu.edu/dm_v6_1_download.shtml (accessed on 25 November 2022)). ^2^ Protein profile information from the ExPASy-ProtParam tool (http://web.expasy.org/protparam/ (accessed on 25 November 2022)). ^3^ Subcellular location information was predicted using the Cell-Ploc (http://www.csbio.sjtu.edu.cn/bioinf/Cell-PLoc/ (accessed on 25 November 2022)) and WoLF PSORT II (https://www.genscript.com/wolf-psort.html (accessed on 25 November 2022)) websites.

**Table 2 ijms-24-11078-t002:** List of the conserved motifs of the StGPXL proteins.

Motif	Length	Amino Acid Sequence
Motif1	50	EILAFPCNQFGAQEPGSNEEIQQFVCTRFKAEFPIFDKIDVNGENAAPLY
Motif2	50	YDFTVKDAKGNDVDLSIYKGKVLLIVNVASKCGLTBSNYTELNQLYEKYK
Motif3	41	KFLKSSKGGFLGDAIKWNFAKFLVDKEGKVVDRYYPTTSPL
Motif4	9	IEKDIKKLL

## Data Availability

Not applicable.

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
