# Peer review of "Genome-Wide Analysis and Expression Profiling of the Glutathione Peroxidase-like Enzyme Gene Family in Solanum tuberosum"

_ijms, 2023, doi:10.3390/ijms241311078_

Round 1

Reviewer 1 Report

Dear Authors,

I have an opportunity to review manuscript entitled: “Genome-Wide Analysis and Expression Profiling of the Glutathione Peroxidase Gene Family in Solanum tuberosum” submitted to IJMS MDPI;

Authors concentrated on potato’s glutathione peroxidase StGPX -identification as well as expression profiles;

Glutathione peroxidases (GPXs) comes from biochemical enzymes classification [E.C.1.11.1.9, E.C. 1.11.1.12, E.C. 1.11.1.15] i.e. phospholipid-hydroperoxide glutathione peroxidase or thioredoxin-dependent glutathione peroxidase, therefore, my question is, which group this article replies to ?

Introduction is sufficient background for the readers, but one aspect should be explained:

Plant GPXs are closely related to mammalian ones than to fungal GPXs; thus, they are named “glutathione peroxidase-like enzymes”, what was in Arabidopsis ‘named’ or ‘ written’ GPXLs- therefore, I warmly suggested using these nomenclature in the text;

Introduction line 48- please add citation about A.th peroxidases;

Is cellular localization added only based on prediction data or subcellular localization were confirmed ?

Figure 1 should be a bit improve to be more readable for the reader;

Line 288 and others- Authors stated “In order to study the expression of potato GPX genes in various tissues and organs during growth and development” – and we have: potato GPX genes in callus, carpels, flowers, leaves, petals, petioles, roots, shoots and more – it was not plant’s tissues ! In the whole manuscript this factual error were repeated!- please correct it;

Why to the expression heat map analyses Authors use Phytophtora (figure 12) as a biotic stress factor and to expression analyses figure 13 Ralstonia was as biotic factor ?

When Authors use “P. Infestans” – should be “infestans”;

Results Part 2.9- The heatmap is very informative part of the figures- but what does it mean “highly expressed” and…

What does it mean“in small amounts in petioles, stems, and stolons”;

Figure caption and chart instead of GMI100 it should be the name of the pathogen;

I think it will be nice to add some schematic summarizing figure/diagram- what S.tuberosum glutathione peroxidases were especially induced/involved in abiotic stress what in biotic stress treatment with Ralstonia or Phytophtora;

Sincerely

Moderate correction should be done after manuscript improvement adding;

Reviewer 2 Report

The authors of the manuscript titled “Genome-Wide Analysis and Expression Profiling of the Glutathione Peroxidase Gene Family in Solanum tuberosum” The current experiment aimed to investigate the potato GPX gene family and identify eight distinct potatoes GPX genes.

General comments

Overall, the study is well-designed and presented in a good way and I accept it in its present form.

Author Response

Dear Reviewer,

Thanks very much for taking your time to review this manuscript. We really appreciate all your generous comments! If you have any questions, please contact us without hesitation.

Yours sincerely,

Shenglan Wang

Reviewer 3 Report

The manuscript submitted by Shenglan Wang , Xinxin Sun , Xinyue Miao , Fangyu Mo , Tong Liu , Yue Chen is a fundumental study elucidating the role of individual potato glutathione peroxidases in stress tolerance. The study is correctly planned, well-executed, the material is clearly visualized. Undoubtedly, the manuscript is of great interest and should be published in a journal. The discussion and conclusion generally correspond to the results and demonstrate the role of individual glutathione peroxidases; however, their role would be more fully described if the authors correlated the observed effects of stressors on the expression of glutathione peroxidases with plant organs and the localization of glutathione peroxidases in different cellular compartments. This would make the manuscript more important.

The presented manuscript is an interesting and important research for understanding the molecular basis of plant resistance.

Author Response

Dear Reviewer,

Thanks very much for taking your time to review this manuscript. We really appreciate all your generous comments and suggestions! If you have any questions, please contact us without hesitation.

Yours sincerely,

Shenglan Wang

Round 2

Reviewer 1 Report

Authors significantly improved manuscript and almost all suggestions and explanations are added.

I warn also against some statements like: highly expressed or low presence- it should be added compared to ...what kind of combination/ what kind of control or reference gene;

I warmly suggest in further research besides expectation/prediction of localisation to add some important and interesting conducted moleculaes subcellular localisation, it makes the results even more informative.